# A Bayesian Network Approach to Lung Cancer Screening: Assessing the Impact of Data Quantity, Quality, and the Combination of Data from Danish Electronic Health Records

**DOI:** 10.3390/cancers16233989

**Published:** 2024-11-28

**Authors:** Florian van Daalen, Margrethe Høstgaard Bang Henriksen, Torben Frøstrup Hansen, Lars Henrik Jensen, Claus Lohman Brasen, Ole Hilberg, Martin Ask Klausholt Andersen, Elise Humerfelt, Leonard Wee, Inigo Bermejo

**Affiliations:** 1Department of Radiation Oncology (MAASTRO), GROW School for Oncology and Reproduction, Maastricht University Medical Centre+, 6229 HX Maastricht, The Netherlands; florian.vandaalen@maastro.nl (F.v.D.);; 2Department of Oncology, Vejle University Hospital, 7100 Vejle, Denmark; torben.hansen@rsyd.dk (T.F.H.); lars.henrik.jensen@rsyd.dk (L.H.J.); 3Institute of Regional Health Research, University of Southern Denmark, 5230 Odense, Denmark; claus.lohman.brasen@rsyd.dk (C.L.B.); ole.hilberg@rsyd.dk (O.H.); 4Department of Biochemistry and Immunology, Vejle University Hospital, 7100 Vejle, Denmark; 5Department of Internal Medicine, Vejle University Hospital, 7100 Vejle, Denmark; 6The Faculty of Health Sciences, University of Southern Denmark, 5230 Odense, Denmark; 7Data Science Institute (DSI), Hasselt University, 3500 Hasselt, Belgium

**Keywords:** lung cancer, bayesian networks, prediction models, screening, early detection, missing data, risk stratification

## Abstract

This study developed and evaluated Bayesian Network models for lung cancer risk prediction using a decade of data from 38,944 high-risk individuals in Denmark. The models were trained and validated on datasets with varying sizes and levels of missing data to reflect real-world screening scenarios. The results showed that a model trained on a small, complete dataset (AUC 0.78) performed similarly on a larger dataset with 21% missing data (AUC 0.78), but performance decreased when 39% of data were missing (AUC 0.67). The laboratory results and smoking data were the most informative variables, significantly outperforming models based only on age and smoking status (AUC 0.70). These findings suggest that BN models can maintain strong predictive performance despite incomplete data and highlight the value of including standard laboratory results in future LC screening programs.

## 1. Introduction

Lung cancer (LC) is the most common cancer globally. It ranks as the most common cancer in men and the second most common in women worldwide, with approximately 2.5 million new cases and 1.8 million deaths in 2022 [1,2,3]. The potential to reduce mortality through early detection via low-dose computed tomography (LDCT) screening has been in focus since the pivotal results from the National Lung Screening Trial (NLST) in the US and the Dutch–Belgian Randomized Lung Cancer Screening Trial (NELSON) [4,5]. These landmark studies have driven numerous local pilot programs and the gradual implementation of LC screening across various countries [6,7,8,9,10,11,12,13,14,15,16]. Presently, the United States is the only country with nationwide LC screening, guided by the U.S. Preventive Services Task Force (USPSTF) recommendations. These guidelines advise LDCT screening for individuals aged 50–80 who are current smokers or have quit within the past 15 years and have a 20-pack–year smoking history [17]. Similar to the USPSTF criteria, most screening trials focus on dichotomized risk factors such as age and smoking intensity. While effective in identifying many high-risk individuals, this approach misses a considerable number of LC cases, particularly among those with atypical risk profiles. For instance, under the current USPSTF criteria, only 68% of LC cases in the United States are detected [18].

Several individualized risk models have shown superior performance over these traditional selection criteria. Among the most recognized and widely integrated are the Liverpool Lung Project (LLP) model [19] and the Prostate, Lung, Colorectal, and Ovarian (PLCOm2012) model [20,21,22]. These models have been employed to select participants based on individualized risk scores in the UK Lung Cancer Screening (UKLS) trial and the ongoing 4-in-the-Lung-Run trial [6,8].

While logistic regression models like LLP and PLCOm2012 are interpretable and well-suited for certain data structures, prediction models based on artificial intelligence (AI) and machine learning (ML) methods offer additional advantages. These models excel in handling complex, high-dimensional, and non-linear data, aligning with the growing use of electronic health record data in predictive modeling [23].

Bayesian networks (BNs), introduced in the 1980s as a subfield of AI, are graphical models that represent probabilistic dependencies among variables. They use directed acyclic graphs (DAGs) to represent the structure of these dependencies. Each variable in a BN is associated with a set of conditional probability distributions, which serve as the parameters that define how the variables interact and influence each other [24].

BNs offer several advantages over other ML methods. Unlike many ML models that depend on imputation or complete-case analysis, BNs handle missing data within their probabilistic framework, estimating the most likely values based on observed variables. A major advantage of BNs is their ability to manage missing data not only during the training phase but also during the inference or classification phase. This capability, along with their strength in modeling complex relationships, has led to growing interest in their application within medical healthcare, especially for disease diagnosis, treatment planning, and decision-making. The scope of BN applications is broad, with notable usage in areas such as cardiology, oncology, psychiatry, and pulmonology [25,26]. The intuitive nature of DAGs also allows for the incorporation of expert knowledge, making BNs particularly relevant in LC screening scenarios.

We previously investigated the risk of LC based on smoking and laboratory results from high-risk individuals suspected of having LC in Southern Denmark [27]. From this dataset, we developed ML methods and BN models capable of predicting LC risk [28,29]. In this study, we expand our analysis to incorporate additional data sources from a larger cohort, resulting in datasets with varying levels of missing data that better reflect real-world conditions. Our study aims to assess how data quantity and quality affect the resulting models and their validation in different combinations of data types from Danish electronic health records and registry data. Specifically, we aim to address the following objectives:

Can a model trained on high-quality, complete data still perform well when validated on data of lower quality?

How does model performance vary with different levels of data completeness and dataset sizes?

Which combination of risk factors yields the best performance?

## 2. Materials and Methods

### 2.1. Study Cohort and Data Sources

The study cohort comprised patients evaluated at the LC fast-track clinics in the Region of Southern Denmark from 1 January 2009 to 31 December 2018. Details on the definition of the study cohort and data sources are outlined in related work [27]. In the final cohort of 38,944 individuals examined for suspected LC, 29% were diagnosed with LC (Figure 1). A broad range of data was collected on this cohort, categorized into four datasets based on data availability:

Comorbidity Dataset: This dataset contains binary information on the international classification of diseases—10th revision (ICD-10) codes obtained from a hospital level, the presence of prescribed medications indicated by anatomic therapeutic chemical (ATC) codes, and the number of consultations and C-reactive protein (CRP) rapid tests conducted in general practice. ICD-10 codes were registered if they appeared within two years prior to the LC fast-track examination, while the interval for ATC codes and general practice data was six months. These data were available for the entire population and were filtered to a subset of ICD-10 and ATC codes that field experts identified as being potentially associated with LC risk [27].

Laboratory Dataset: This dataset includes the results of 20 common laboratory analyses performed at the LC fast-track clinics. These analyses were included if conducted within the period of four weeks before the LC examination until two weeks after. This was performed in order to reflect the laboratory status at the time of the LC examination.

Smoking Dataset: This dataset provides binary information on smoking status, categorized as either never smoker or current/former smoker. This information was derived from free-text data in subfields of the electronic health records, which were manually annotated when available. The absence of data in this subfield limited this dataset to a subset of the entire population [30].

Symptom Dataset: This dataset contains manually annotated data from free text regarding the presence of the most common symptoms recorded in the primary journal of the LC examination. Additionally, it includes information on genetic predispositions to LC (parents or siblings with LC) and relevant exposures, such as radon or asbestos, based on the same records. The primary journals were filtered to include entries recorded within four weeks before to two weeks after the LC examination date. To reduce the burden of manual annotation, we focused on annotating data for a subset of individuals with complete information from the previously described datasets. This reduced cohort has previously been compared to the remaining individuals with missing data and showed overall similarity. However, the reduced cohort had a lower rate of comorbidities among LC patients and exhibited fewer differences in medication prescriptions compared to the remaining cohort [27].

### 2.2. Experimental Setup

#### 2.2.1. Discretization

Before constructing the BN models, continuous variables were binned using the minimum description length (MDL) strategy [31]. This strategy seeks to find the optimal number of bins for the continuous laboratory results. The optimal number of bins is described by a model that offers the shortest overall description of the data, balancing model complexity (avoids overfitting) and accuracy (avoids underfitting). In initial exploratory experiments, we compared the MDL discretization method with the standard clinical reference intervals provided by the laboratory departments, which are typically based on 95% confidence intervals. We observed no significant differences between the two approaches and, for the sake of simplicity, chose to use the MDL strategy in the final experiments.

#### 2.2.2. Model Development

The development of the BN models involved two key phases: structure learning and parameter learning. For structure learning, we employed the K2 algorithm introduced by Cooper et al. in 1992 [32]. This algorithm uses a greedy search strategy to identify the most suitable structure for the DAG. It iteratively adds one variable at a time, aiming to maximize the scoring function, which reflects the likelihood of the data given the network structure. This process continues until no further improvement in the score is observed or the maximum number of parent nodes per variable is reached, which was tested with 1–10 parent nodes in this case. The data used for structure learning needed to be complete, so missing continuous variables were imputed with the mean while missing discrete variables were imputed with the mode. The best structure was selected based on its area under the ROC curve (AUC), as estimated using 10-fold cross-validation.

In the parameter learning phase, we learned the conditional probability tables using the Expectation–Maximization (EM) algorithm [33]. The EM algorithm handles missing values in two steps: the expectation step, where it estimates the missing data based on the observed data, and the maximization step, where it adjusts the parameters to maximize the likelihood of the observed data. These steps are repeated until the parameter estimates stabilize.

We conducted preliminary exploratory experiments comparing the K2 algorithm with expert-drawn graphs and found no significant differences between them. For simplicity, we decided to use the K2 algorithm for the experiments described in this article.

#### 2.2.3. Division of Study Cohort

To investigate model performance with varying dataset sizes, completeness levels, and attributes, the study cohort was divided into subsets. Figure 2 displays these subsets, Datasets A to D, derived based on the four data categories: Dataset A includes individuals holding only comorbidity data; Dataset B includes individuals with comorbidity and laboratory results; Dataset C holds individuals with comorbidity, lab, and smoking data; and Dataset D individuals encompassing all four categories.

Datasets A–D were combined to reflect a real-world distribution of missing data within datasets, where certain groups of individuals have complete data, while others lack information in some categories, such as smoking habits or symptoms. While Dataset D represents a small but nearly complete dataset with only 2% missing data, the other dataset combinations have higher rates of missing data: Dataset CD has 16% missing data, Dataset BCD has 21% missing data, and Dataset ABCD has 39% missing data. It should be noted that in the less complete datasets, the missing data levels include variables that are completely missing. General variables such as sex and age were included in all developed models.

### 2.3. Evaluation Setup

#### 2.3.1. Evaluation 1

Models were trained on the records present in the four combinations of increasingly complete datasets (ABCD/BCD/CD/D), as described in Figure 2. Additionally, each of these four combinations was trained using all possible sub-combinations of the various data categories. This results in 15 different models, which can be found in Figure 3. One example is a model trained on dataset ABCD, using only the comorbidity-related variables.

Validation involved a combination of 10-fold cross-validation on overlapping datasets and external validation on non-overlapping datasets. For example, a model trained on dataset D was evaluated using cross-validation within dataset D. In addition, the model trained on dataset D was externally validated on dataset A. When larger datasets were used for validation, the results from both validation methods were integrated. For instance, the performance of a model trained on dataset D and validated on dataset ABCD was assessed by combining the cross-validation results from dataset D with the external validation results from datasets A, B, and C.

#### 2.3.2. Evaluation 2

To determine the optimal set of variables overall, we used the most complete dataset (dataset D) for both training and validation using 10-fold cross-validation. We evaluated the performance using their AUC score of the 15 potential combinations of the four data categories shown in Figure 3. By comparing the same dataset, the cohort size remained consistent, ensuring that any performance improvements were solely due to the combination of variables rather than an increase in cohort size.

### 2.4. Statistical Analyses

Baseline characteristics were described using the median and interquartile range (IQR) for continuous variables and number and percentage for categorical variables. The validation of the experiments was conducted using AUCs, along with 95% confidence intervals (CIs) calculated using a standard normal distribution. Discrimination was assessed through the AUCs, and the true positive rate (TPR/sensitivity) and true negative rate (TNR, specificity) were evaluated at the default probability cut-off of 0.5. All experiments were conducted using the WEKA framework version 3.8 [34].

## 3. Results

### 3.1. Baseline Characteristics

Table 1 presents the baseline characteristics for each dataset category. The LC group was older and had a higher proportion of females compared to the non-LC group, with ages of 70 years (IQR 63–77) versus 67 years (IQR 56–75) and 48.4% versus 45.3% females, respectively (*p* < 0.001 for both). Both groups commonly had comorbidities such as other malignancies (13.3%), chronic pulmonary disease (12.4%), and pneumonia (10.5%). However, LC patients were significantly more likely to have metastatic solid tumors, cerebrovascular disease, and peripheral vascular disease, whereas non-LC patients had a higher prevalence of other malignancies. Antibiotics were the most frequently prescribed medication in both groups, with no variation across groups (43.9%). Other medication categories were more frequently prescribed to LC patients.

In laboratory results, most median values fell within clinical standard reference intervals, which were based on 95% CIs. Nevertheless, LC patients showed significantly elevated levels of white blood cells (leukocytes, neutrophils, monocytes), platelets, calcium, CRP, lactate dehydrogenase (LDH), and alkaline phosphatase compared to non-LC patients. Conversely, LC patients had lower levels of hemoglobin, eosinophils, lymphocytes, albumin, alanine aminotransferase (ALAT), creatinine, and sodium. The proportion of current or former smokers was significantly higher among LC patients (92%) compared to non-LC patients (69%, *p* < 0.001). The most common symptoms in both groups included cough (53.4%), dyspnea (36.3%), weight loss (25.2%), fatigue (19.9%), and hemoptysis (16.2%). Weight loss, fatigue, back pain, and other pain symptoms were more prevalent among LC patients, while hemoptysis and fever were more common in the non-LC cohort (*p* < 0.001 for all). For further details on baseline characteristics, please refer to the related literature [27].

### 3.2. Performance Assessment

#### 3.2.1. Evaluation 1

Table 2 presents the AUC values for models trained on datasets ABCD, BCD, CD, and D, with validation conducted on datasets A, B, C, D, BCD, CD, and ABCD. For each combination, the 15 possible combinations of the four data types were evaluated, but for simplicity, only the best-performing combination is shown.

Model performance varied significantly, ranging from an AUC of 0.60 (95% CI 0.59–0.61) for the worst-performing model to 0.79 (95% CI 0.75–0.83) for the best-performing model. The top-performing model was trained using dataset BCD and validated on dataset B using cross-validation, incorporating data on comorbidities, laboratory results, and smoking history. This dataset included 14,957 individuals and had 21% missing data. There was no significant difference between this model and the second-best model when considering the overlap in confidence intervals. The second-best model achieved an AUC of 0.78 (95% CI 0.75–0.82) and was trained and validated on the smaller, nearly complete dataset D, utilizing all four types of data variables. When tested on larger datasets with higher rates of missing data, its performance remained relatively stable, with AUC values slightly decreasing to 0.77 (95% CI 0.74–0.80) on dataset CD (16% missing data) and 0.78 (95% CI 0.75–0.80) on dataset BCD (21% missing data). However, its performance declined significantly to an AUC of 0.67 (95% CI 0.66–0.68) when validated on dataset ABCD, which had 39% missing data. Overall, all models performed poorly when validated on dataset ABCD, particularly on dataset A alone, regardless of the training set used.

#### 3.2.2. Evaluation 2

Figure 4 illustrates the performance comparison across the 15 dataset combinations, using dataset D for training and validation. The top-performing model, which utilized all four types of data, achieved an AUC of 0.783 (95% CI 0.748–0.818). This model, previously identified as the second-best overall, showed slightly better performance compared to the model that used only laboratory and smoking data (AUC 0.761, 95% CI 0.727–0.795), though there was considerable overlap in confidence intervals. Both of these models significantly outperformed the one based solely on smoking status, along with age and gender, which had an AUC of 0.702 (95% CI 0.673–0.731). Overall, models that included laboratory and smoking data achieved the best performance, while comorbidities and symptoms had the least impact on model performance.

## 4. Discussion

### 4.1. Summary of Results

In this study, we developed BNs to predict the risk of lung cancer (LC) among a high-risk population in southern Denmark based on a decade of data. We analyzed data from 38,944 individuals, categorized into four types: comorbidity data, laboratory results, smoking history, and symptom-related data. To evaluate model performance across varying data completeness and sizes, the dataset was divided into smaller subsets based on the structure of the missing data.

We found that a relatively small high-quality training dataset could produce acceptable performance when validated on a dataset of lower quality. This was demonstrated by both the top-performing model, trained on dataset BCD, and the second-best model, trained on dataset D. Model performance remained largely stable until tested on a dataset containing 39% missing values (dataset ABCD). The best results were achieved by incorporating all four types of data (comorbidities, laboratory, smoking, and symptoms). Overall, laboratory and smoking data had the greatest impact on model performance, while symptom-related data had the least influence.

### 4.2. Interpretation and Comparison

We demonstrated that a model trained on high-quality data maintains strong performance even when validated against datasets with higher rates of missing information. This was true for both the top-performing model trained on dataset BCD and validated on dataset B, as well as the second-best model trained on dataset D, which yielded nearly identical results when validated on datasets CD, BCD, and B. This indicates that including lower-quality data does not significantly impair the model’s performance and can still offer valuable insights. For example, a model built from comorbidity, laboratory, smoking, and symptoms data on a small cohort (dataset D) remains consistently effective when applied to larger cohorts with partially missing data for smoking and symptoms (dataset BCD).

This ability to handle missing data is advantageous for screening, as it allows a model to be trained on a bigger cohort, which includes individuals with some missing data, without immediately harming model performance. This approach enables a larger proportion of individuals to be included in the training data without the need to exclude or impute missing data, which can introduce bias or errors, especially if the missing data are not missing at random. For instance, different subpopulations may have a different probability of having a specific laboratory test taken, resulting in different rates of missingness for this variable. Imputing this test based on the global population could introduce bias. BNs diminish these biases by using a probabilistic framework that accounts for the different subpopulations [24].

The model’s ability to handle missing data makes it scalable to various populations, including those in resource-limited settings where comprehensive data collection, such as smoking history or symptoms, may be difficult. For example, even when relying solely on data from ICD-10 comorbidity diagnoses and laboratory results due to recruitment challenges, the model still achieves an acceptable AUC of 0.77 when trained on dataset D and validated on dataset B.

This ability to handle missing values shows the advantages of using a BN during the training phase. In addition to this, it brings a similar benefit when the finalized model is utilized in clinical practice. Since the model can handle incomplete records, it becomes possible to use the model to screen a new patient without having to apply the full set of medical tests. This means that expensive, invasive, time-consuming, or otherwise inconvenient data to collect could potentially be skipped without compromising the accuracy of the model. This is especially relevant in screening scenarios as this reduces the barrier to entry, potentially allowing more individuals to be screened at a cheaper cost and with less discomfort.

However, when validated on dataset ABCD, which has 39% missing data, the model’s performance drops significantly, indicating a threshold for the amount of missing data the model can effectively handle. This effect may be compounded by the fact that dataset ABCD introduces a large number of records with missing lab values, which appear to be an important variable in our model. However, it should be noted that the drop in performance is present regardless of which set of attributes was used. It is well-known from the literature that although BNs are capable of managing missing data, their performance deteriorates as the quality of missing data decreases [35,36]. The exact rate or threshold of deterioration can vary depending on factors such as the dataset, model complexity, and the pattern of missing data. In cases where data are not missing at random—such as in this study, where entire categories of data were missing (e.g., all comorbidity or all lab values)—the impact on model performance is more pronounced compared to situations where data are missing completely at random [35,37].

Additionally, performance declines when validating on dataset A alone, suggesting that the comorbidity dataset by itself may not provide sufficient information for LC detection. This could be due to the fact that a significant proportion of patients in both groups had any of the included comorbidities: 62% of the LC cohort and 65% of the non-LC cohort. Furthermore, the comorbidity dataset included data on general practitioner visits and CRP measurements, but these variables did not significantly differentiate between LC and non-LC patients. Consequently, their presence likely diminished the dataset’s overall significance. Despite a higher overall rate of prescription medication among LC patients, this difference did not substantially enhance the predictive power when combined with variables related to ICD-10 codes and general practice.

The performance of four combinations of variables was compared, with the best results achieved by using all four data types. This combination of all four data types was comparable to the models that only used laboratory and smoking data, suggesting that adding comorbidity and symptoms data can only slightly enhance performance. Interestingly, both these combinations surpassed the conventionally used LC screening criteria, which rely solely on smoking status (along with age and sex). This underlines the potential of using other data sources in the selection of eligible individuals. Laboratory and smoking data have already been utilized by the members of the same study group to create ML and BN models [28,29]. In those models, dataset C, comprising 9940 individuals, was used with a focus on smoking and laboratory data, as initial exploration highlighted their significance in distinguishing LC from non-LC patients. The DES model developed by Flyckt et al. achieved a performance of 0.77, while the BN model by Henriksen et al. reached a similar performance of 0.76 on the same dataset. In this current study, dataset C was not used alone for model training, so a direct comparison cannot be made. However, the results indicate that the performance of both previous models can be matched or even surpassed by incorporating additional data sources (such as comorbidity and symptoms data) and/or expanding to a larger dataset despite its higher rate of missing data.

The PLCOm2012 model, which is frequently used, has demonstrated superior performance compared to the NLST eligibility criteria, with AUC values ranging from 0.76 to 0.81 in validation studies [38,39,40]. Although these results are similar to those found in this paper, a direct comparison is challenging. This is because the PLCOm2012 model was developed and validated using screening populations with a lower incidence of LC than in the current study. Additionally, the PLCOm2012 model is designed to predict 6-year risk, whereas this study focuses on predicting risk at the time of examination or diagnosis, which is more about detection. Despite these differences, our findings are noteworthy given that all individuals in this study are considered high-risk, and the control group is not as healthy as those in typical screening populations. This makes it more difficult for the model to discriminate LC patients from non-LC individuals.

Future research will involve analyzing data from COPD outpatients who are at moderate risk for LC compared to the high-risk patients in this study. Additionally, with a Danish LC screening trial in the planning phase, exploring the development or validation of BNs for this cohort could be valuable. This research could help assess whether these models can exceed the performance of the criteria used in the forthcoming Danish screening program.

### 4.3. Methodological Considerations

The models developed in this study utilized a large dataset spanning a decade and covering an entire region of Denmark. This dataset stands out due to its integration of various data sources, including manual annotations on symptoms, dispositions, exposures, and smoking, from relatively large cohorts—a data collection approach that is quite rare. The combination and diversity of these data sources offer a comprehensive view of the distributions and risk factors for LC within this high-risk population. The experimental design, which combined datasets to simulate varying degrees of missing data, mirrors real-life scenarios where some patients might lack information on smoking and symptoms or may not have undergone laboratory tests. This approach provides a more realistic depiction compared to earlier studies that used artificially introduced missing values that were completely random [29].

Despite its strengths, this study has several limitations that warrant consideration. The analyses rely exclusively on retrospective data, which introduces several biases due to the constraints of using variables collected at specific times. The most critical limitation is selection bias, as the dataset primarily includes detailed information on individuals who have visited hospital settings, with limited representation of the broader population eligible for screening. This focus on high-risk patients may result in a model that performs poorly when applied to less severe cases or healthier individuals. Consequently, while the dataset is valuable for certain analyses, it is suboptimal for LC screening, which requires a dataset that better represents the general population.

Comorbidity data is also biased toward hospital-recorded diagnoses, leading to a high proportion of patients being classified as having no comorbidities. Incorporating data from general practice, such as the International Classification of Primary Care (ICPC) codes that include symptoms and diagnostic information at the population level, could help address this limitation. Furthermore, using symptom data sourced from the general population might enhance the model’s ability to distinguish between LC and non-LC patients, as non-LC individuals would likely exhibit fewer symptoms. Other omitted variables, such as socioeconomic status, may also influence both LC risk and smoking status. Including such data could introduce more variation in risk and improve model discrimination and performance. However, we chose not to include socioeconomic data, as it is centralized at Statistics Denmark and unavailable for real-time use in clinical settings. Instead, we prioritized data sources that are accessible for everyday clinical practice.

Another limitation relates to information bias and the quality of particularly smoking status, which is often inconsistently recorded. A more detailed smoking dataset that includes packs–years and years since quitting could enable more precise predictions and a fairer comparison to current screening criteria. However, the aim of this study was to analyze real-world data, acknowledge its incompleteness and variability, and explore how model performance changes when validated across different subgroups within the dataset.

A more general limitation is linked to the experimental setup, where we analyzed pooled data without assessing individual variables within these datasets. For example, the optimal performance might involve combining COPD, three lab tests, smoking, and age, but our results suggest that the significance of individual variables like COPD could be reduced by the lack of significance from other variables in the comorbidity dataset. Certain medication prescriptions may become more prominent when assessed independently. This approach was chosen to simplify the experimental setup, given the complexity arising from multiple subpopulations with missing data and the inclusion of four different datasets, but will be explored further in future studies.

## 5. Conclusions

Our findings demonstrate that it is feasible to develop an LC prediction model using a high-quality dataset and achieve satisfactory performance even when validating with lower-quality data with up to 39% missing data. The most effective variable combination was achieved by integrating all types of data, with smoking and laboratory results proving to be particularly informative. These results are relevant for LC screening scenarios, where data are often disjointed and difficult to obtain. Future research should focus on validating the model in a broader population, such as COPD outpatients. Furthermore, its clinical potential to improve screening efficiency and optimize resource utilization should be assessed, emphasizing analyses of specific feature importance rather than limiting the focus to the broader data groupings explored in this study. Finally, further exploration of models that can handle missing data is crucial to identifying the most effective strategies for improving prediction accuracy in real-world settings.

## Figures and Tables

**Figure 1 cancers-16-03989-f001:**
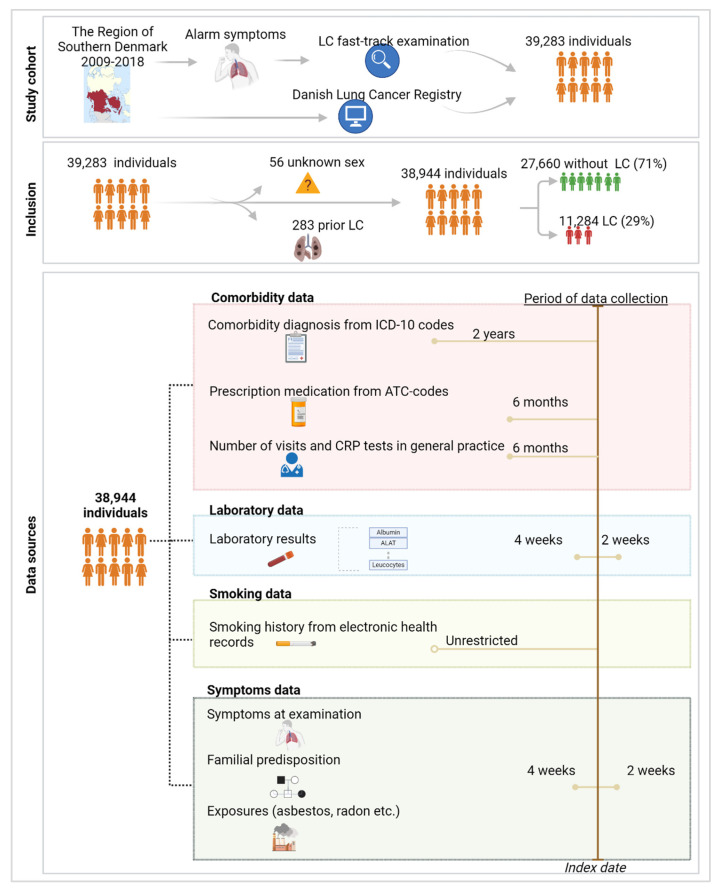
Study cohort, inclusion criteria, and data collection. The data sources included in the study were categorized into comorbidity data, laboratory results, smoking history, and symptoms data. Comorbidity data encompassed information on ICD-10 codes, prescription medications, the number of visits, and quick tests performed in general practice. Laboratory results consisted of 20 different analyses. Smoking history provided detailed records of smoking habits in binary format, while symptoms data included information on common symptoms, familial predispositions, and relevant exposures to LC. These data were collected for specific periods leading up to the date of inclusion, referred to as the index date, and are depicted by the bars on the right side of the image. Created with Biorender.com.

**Figure 2 cancers-16-03989-f002:**
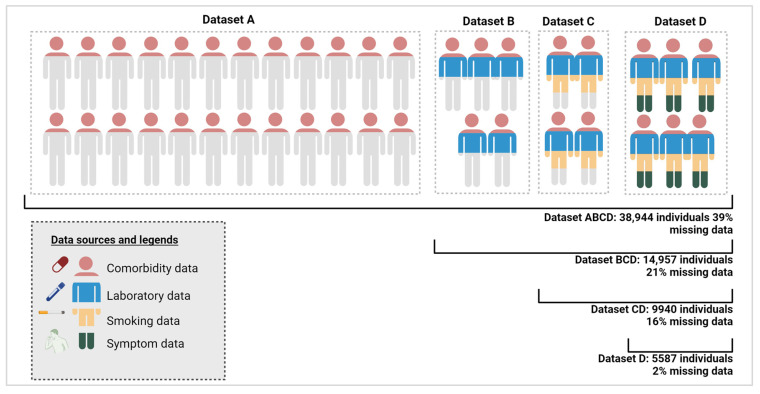
Datasets A–D and the combination of these with increasing degree of missing data. The color code of the individuals reflects the type of data available on this subset from the four combinations of data: comorbidity, laboratory, smoking, and symptoms data. Created with Biorender.com.

**Figure 3 cancers-16-03989-f003:**
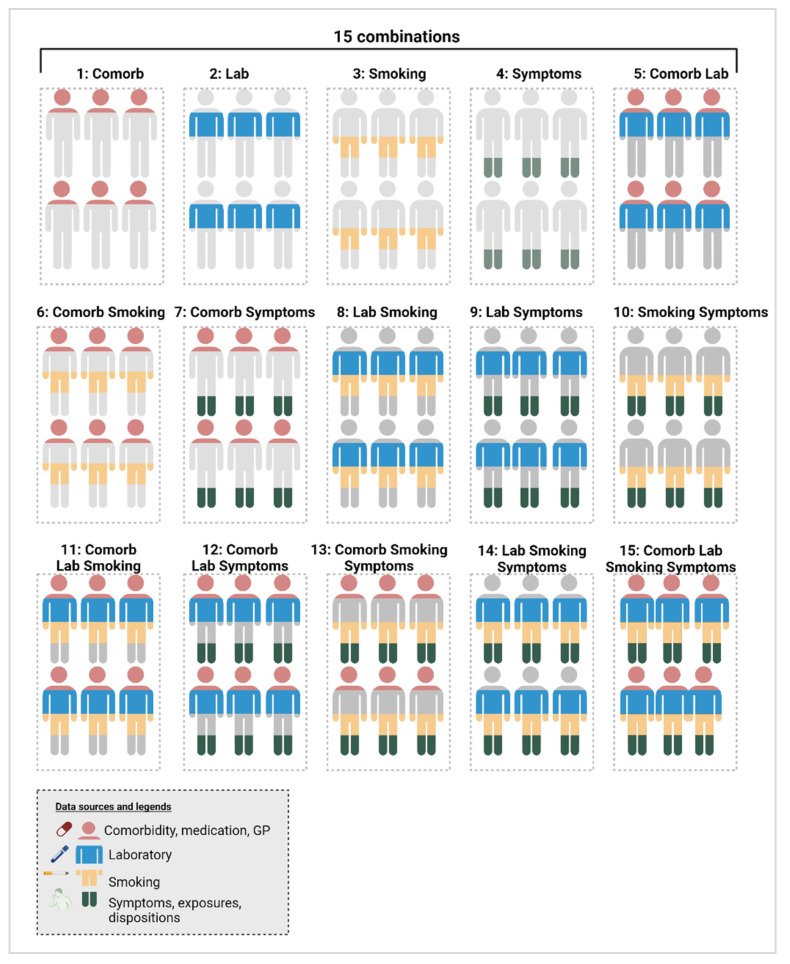
The 15 models trained and validated on dataset D, comparing different combinations of the four data types: comorbidity, laboratory results, smoking history, and symptoms data. Created with Biorender.com, https://www.biorender.com, accessed on 5 August 2024).

**Figure 4 cancers-16-03989-f004:**
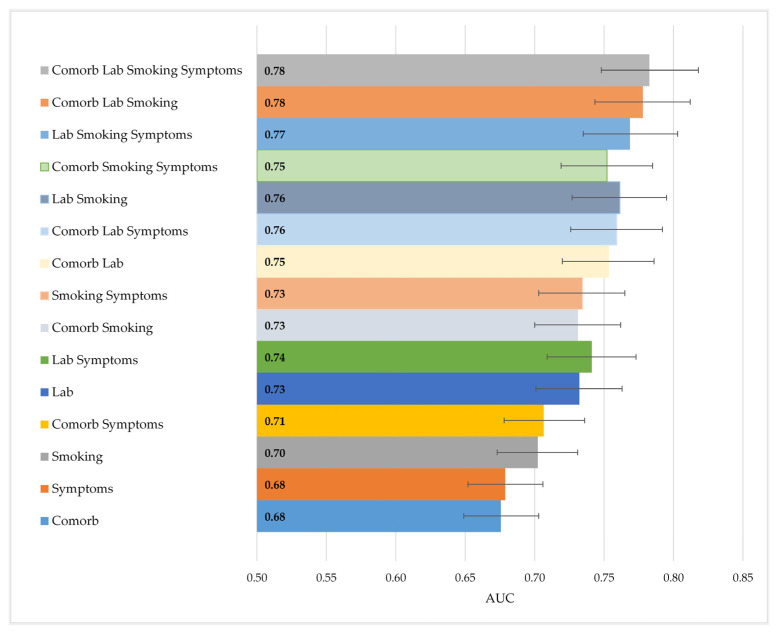
Comparison of AUCs when combining the four types of data in models trained and validated on dataset 4 (5587 individuals). The 95% confidence intervals are indicated with bars.

**Table 1 cancers-16-03989-t001:** Baseline characteristics for the non-LC and LC groups, including demographic data as well as information from the comorbidity, laboratory, smoking, and symptoms datasets. Categorical or binary data are expressed as fractions and compared using chi-squared tests, while continuous variables are shown as medians with interquartile ranges and *p*-values calculated using Wilcoxon signed-rank tests.

Demography	Non-LC	LC	*p*-Value
**Total, no. (%)**	**27,660 (100)**	**11,284 (100)**	
Age, median (IQR)	67 (56–75)	70 (63–77)	<0.001
Females, no. (%)	12,515 (45.3)	5461 (48.4)	<0.001
**LC stage, no. (%)**			
I		2.001 (17.7)	
II		914 (8.1)	
III		2242 (19.9)	
IV		5440 (48.2)	
Unknown		687 (6.1)	
**Comorbidity dataset**	**Non-LC**	**LC**	***p*-value**
**Total, no. (%)**	**27,660 (100)**	**11,284 (100)**	
Myocardial infarction	454 (1.6)	225 (2.0)	0.02
Congestive cardiac failure	198 (0.7)	69 (0.61)	0.26
Peripheral vascular disease	828 (3.0)	555 (4.9)	<0.001
Cerebrovascular disease	915 (3.3)	525 (4.7)	<0.001
Dementia	200 (0.7)	77 (0.7)	0.67
Chronic pulmonary disease	3379 (12.2)	1429 (12.7)	0.22
Rheumatological disease	533 (1.9)	228 (2.0)	0.54
Liver disease	198 (0.7)	65 (0.6)	0.13
Diabetes mellitus	1245 (4.5)	566 (5.0)	0.03
Hemiplegia or paraplegia	30 (0.1)	18 (0.2)	0.19
Renal disease	522 (1.9)	179 (1.6)	0.05
Metastatic solid tumor	772 (2.8)	6.5 (5.4)	<0.001
AIDS/HIV infection	19 (0.1)	9 (0.1)	0.94
Pulmonary tuberculosis	48 (0.2)	8 (0.1)	0.02
Sarcoidosis	79 (0.3)	18 (0.2)	0.02
Interstitial lung disease	194 (0.7)	74 (0.7)	0.62
Abscess	157 (0.6)	43 (0.4)	0.02
Pleural disease	725 (2.6)	297 (2.6)	0.95
Pneumonia	2944 (10.6)	1132 (10.0)	0.07
SumCCI, sum (median)	0 (50)	0 (50)	<0.001
Other malignancies	3859 (14.0)	1321 (11.7)	<0.001
Antibiotics	12,130 (43.9)	4954 (43.9)	0.93
COPD inhalations	7028 (25.4)	3490 (30.9)	<0.001
Antihypertensives	9560 (34.6)	4660 (41.3)	<0.001
Glucocorticoids	2770 (10.0)	1534 (13.6)	<0.001
Metformin	1694 (6.1)	837 (7.4)	<0.001
Antidepressants	3960 (14.3)	1838 (16.3)	<0.001
Consultations at GP	3093 (11.2)	932 (8.3)	<0.001
CRP rapid tests at GP	13,053 (47.2)	5275 (46.8)	
**Laboratory data**	**Non-LC**	**LC**	***p*-value**
**Total, no. (%)**	**10,503 (100)**	**4454 (100)**	
B-Hemoglobin, mmol/L	8.7 (8.0–9.3)	8.40 (7.7–9.0)	<0.001
B-Leucocytes, 10^9^/L	7.64 (6.20–9.46)	9.12 (7.43–11.20)	<0.001
B-Neutrophils, 10^9^/L	4.70 (3.58–6.20)	6.10 (4.71–7.95)	<0.001
B-Lymphocytes, 10^9^/L	1.81 (1.39–2.33)	1.74 (1.30–2.27)	<0.001
NLR	2.6 (1.8–3.8)	3.4 (2.4–5.2)	<0.001
B-Monocytes, 10^9^/L	0.65 (0.51–0.84)	0.76 (0.59–0.97)	<0.001
B-Basophils, 10^9^/L	0.04 (0.02–0.06)	0.04 (0.02–0.06)	<0.001
B-Eosinophils, 10^9^/L	0.17 (0.10–0.27)	0.14 (0.07–0.25)	<0.001
B-Platelets, 10^9^/L	272 (223–334)	311 (250–391)	<0.001
P-Albumin, g/L	43 (41–45)	42 (39–44)	<0.001
Total Calcium, mmol/L	2.34 (2.27–2.41)	2.36 (2.29–2.43)	<0.001
P-CRP, mg/L	3.7 (1.4–10.0)	9.9 (3.0–32.0)	<0.001
P-ALAT, U/L	22 (16–31)	18 (13–26)	<0.001
P-LDH, U/L	192 (169–221)	214 (182–257)	<0.001
P-Alkaline phosphatase, U/L	75 (62–92)	83 (68–102)	<0.001
P-Bilirubin-total, μmol/L	7 (6–10)	7 (5–9)	<0.001
P-Amylase (pancreatic), U/L	25 (19–34)	25 (18–34)	0.79
P-INR	1.0 (0.9–1.1)	1.0 (0.9–1.1)	<0.001
P-Creatinine, mmol/L	76 (64–89)	72 (60–87)	<0.001
P-Sodium, mmol/L	140 (138–142)	139 (136–141)	<0.001
P-Potassium, mmol/L	4.0 (3.8–4.3)	4.0 (3.8–4.3)	0.08
**Smoking status**	**Non-LC**	**LC**	** *p* ** **-value**
**Total**	**7435 (100)**	**2505 (100)**	
Never smoker	2288 (30.8)	196 (1.8)	<0.001
Former or current smoker	5147 (69.2)	2309 (92.2)	
**Symptoms, familial predispositions and exposures, no. (%)**	**Non-LC**	**LC**	** *p* ** **-value**
**Total**	**3733 (100)**	**1854 (100)**	
Predispositions	253 (6.8)	167 (9.0)	0.00
Exposures	785 (21.0)	354 (19.1)	0.09
Hemoptysis	694 (18.6)	212 (11.4)	<0.001
Pneumonia	671 (18.0)	303 (16.3)	0.13
Cough	2012 (53.9)	969 (52.3)	0.25
Dyspnoea	1365 (36.6)	663 (35.8)	0.56
Fever	286 (7.2)	81 (4.4)	<0.001
Weight loss	822 (22.0)	584 (31.5)	<0.001
Fatigue	684 (18.3)	428 (23.1)	<0.001
Hot flash	402 (10.8)	177 (9.6)	0.16
Hoarseness	174 (4.7)	92 (5.0)	0.62
Back pain	133 (3.6)	129 (7.0)	<0.001
Other pain	340 (9.1)	250 (13.5)	<0.001
Angina	428 (11.5)	256 (13.8)	0.01
Headache	144 (3.1)	65 (3.5)	0.37
Dizziness	161 (4.3)	96 (5.2)	0.15
Edema	196 (5.3)	108 (5.8)	0.37

**Table 2 cancers-16-03989-t002:** AUC measures and 95% CI obtained from various combinations of datasets A-D used for training and validation. For each dataset, the table includes the results of testing all 15 possible combinations of data types, with the optimal combination highlighted for each case. The two best performing models are highlighted in bold.

Val. Data	Training Data ABCD		Training Data BCD		Training Data CD		Training Data D	
	Variables	AUC (95% CI)	Variables	AUC (95% CI)	Variables	AUC (95% CI)	Variables	AUC (95% CI)
**A**	Comorb Symptoms	0.63 (0.62–0.64)	Comorb Lab	0.62 (0.60–0.63)	Comorb	0.60 (0.59–0.61)	Comorb Smoking Symptoms	0.60 (0.59–0.61)
**B**	Comorb Lab Symptoms	0.78 (0.75–0.82)	Comorb Lab Smoking	**0.79** **(0.75–0.83)**	Comorb Lab	0.78 (0.74–0.81)	Comorb Lab	0.77 (0.73–0.80)
**C**	Lab Smoking	0.72 (0.67–0.76)	Lab Smoking Symptoms	0.72 (0.78–0.77)	Lab Smoking Symptoms	0.73 (0.69–0.78)	Lab Smoking	0.73 (0.58–0.80)
**D**	Comorb Lab Smoking Symptoms	0.75 (0.72–0.79)	Comorb Lab Smoking Symptoms	0.77 (0.73–0.80)	Comorb Lab Smoking	0.77 (0.73–0.80)	Comorb Lab Smoking Symptoms	**0.78** **(0.75–0.82)**
**CD**	Comorb Lab Smoking Symptoms	0.76 (0.73–0.79)	Comorb Lab Smoking Symptoms	0.77 (0.74–0.80)	Comorb Lab Smoking Symptoms	0.78 (0.75–0.81)	Comorb Lab Smoking Symptoms	0.77 (0.74–0.80)
**BCD**	Comorb Lab Smoking Symptoms	0.77 (0.75–0.79)	Comorb Lab Smoking Symptoms	0.78 (0.76–0.80)	Comorb Lab Smoking	0.78 (0.75–0.80)	Comorb Lab Smoking Symptoms	0.78 (0.75–0.80)
**ABCD**	Comorb Lab Smoking Symptoms	0.69 (0.68–0.70)	Comorb Lab Smoking	0.68 (0.67–0.69)	Comorb Lab Smoking	0.66 (0.65–0.67)	Comorb Lab Smoking Symptoms	0.67 (0.66–0.68)

## Data Availability

Data may be shared upon reasonable request.

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
