# Peer review of "A Bayesian Network Approach to Lung Cancer Screening: Assessing the Impact of Data Quantity, Quality, and the Combination of Data from Danish Electronic Health Records"

_cancers, 2024, doi:10.3390/cancers16233989_

Round 1
Reviewer 1 Report
Comments and Suggestions for Authors
The article primarily discusses the Bayesian network approach to lung cancer screening, evaluating the impact of the quantity and quality of data combined from Danish electronic health records. It focuses on the analysis of a large sample of high-risk individuals, aiming to improve lung cancer risk prediction and optimize screening strategies. Additionally, the article addresses issues related to the distribution of missing data and the importance of having complete information on factors such as smoking history and symptoms.
In my personal opinion there are issues that have to be faced:
1) Firstly, the reliance on retrospective data may introduce biases, as the data collected may not encompass all relevant variables or may be incomplete.
2)The presence of missing data in certain subsets could affect the performance of the Bayesian network models, as indicated by the drop in predictive performance when a significant portion of data was missing.
3)The study's findings may be limited in their generalizability, as the data is derived from a specific Danish population, which may not reflect the characteristics of other populations or healthcare systems.
4) The complexity of the Bayesian network models may pose challenges in their practical implementation in clinical settings.
5) It could be valuable to include specific cases to better contextualize the analyzed scenario. Works such as Lung Cancer Detection via Federated Learning,A comparative study of linear type multiple instance learning techniques for detecting COVID-19 by chest X-ray images could provide interesting insights.
Reviewer 2 Report
Comments and Suggestions for Authors
The authors have prepared an important manuscript on the use of Bayesian models in predicting lung cancer in high-risk polpulations based on existing data from patient files.
The population used for the analysis is quite large and the design of the study has considered latest advances in the field of lung cancer screening strategies globally.
The authors have delivered important outcomes that will affect future strategies.
Some minor comments:
- Part of the references, especially those based on your previous work (e.g. 26,27) are not properly reported. Please update and revise.
- Please specify in your conclusion WHICH laboratory exams were important in the model.
Reviewer 3 Report
Comments and Suggestions for Authors
The article presented to me for review shows the potential of artificial intelligence methods, in particular Bayesian networks, in supporting the diagnosis of lung cancer.
Below are my comments and suggestions regarding this article:
1) In the introduction, the authors write about 2.5 million cases of lung cancer detected in 2022. I think it would be valuable to expand this information to include the number of deaths.
2) In chapter 2.1. Study cohort and data sources, the authors present a brief description of the data. In my opinion, it would be worthwhile to prepare more detailed tables presenting this data in the additional materials, containing information such as the attribute value, min, max, mean, standard deviation and histograms of selected features. This will allow for a better understanding of the data on which the experiments were conducted.
3) In the research, the authors mainly focused on the use of Bayesian networks - it would be good to briefly mention the use of this type in the network in research aimed at building intelligent systems used in medicine.
4) In their research, the authors mainly relied on the AUC metric. Additional metrics such as accuracy and f1-score should be added to fully accurately present the results achieved. For the best models, confusion matrices and ROC curves should be prepared. If they would take up too much space in the article, they can be included in the additional materials.
5) Do the authors plan to use classical machine learning methods in the future, e.g. from the field of gradient boosting together with typical methods of filling in missing values ​​such as knn - to compare the achieved results with Bayesian networks?
